# Spatial Transcriptomic Analysis Reveals Associations between Genes and Cellular Topology in Breast and Prostate Cancers

**DOI:** 10.3390/cancers14194856

**Published:** 2022-10-04

**Authors:** Lujain Alsaleh, Chen Li, Justin L. Couetil, Ze Ye, Kun Huang, Jie Zhang, Chao Chen, Travis S. Johnson

**Affiliations:** 1Department of Biostatistics and Health Data Science, Indiana University, Indianapolis, IN 46202, USA; 2Department of Biomedical Informatics, Stony Brook University, Stony Brook, NY 11794, USA; 3Department of Medical and Molecular Genetics, Indiana University, Indianapolis, IN 46202, USA; 4Regenstrief Institute, Indiana University, Indianapolis, IN 46202, USA; 5Melvin and Bren Simon Comprehensive Cancer Center, Indiana University, Indianapolis, IN 46202, USA; 6Indiana Biosciences Research Institute, Indianapolis, IN 46202, USA

**Keywords:** spatial transcriptomics, histopathological images, image analysis, breast cancer, prostate cancer, gene expression, topological data analysis, integrative analysis

## Abstract

**Simple Summary:**

Tissues consist of various cell types in complex spatial arrangements. These tissues require signaling between the cells and structural proteins to maintain their organization. Spatial arrangements or the organization of cells in tissue can be readily studied via imaging analysis of dissected tissue. In cancer tissues, there is, additionally, signaling between immune cells and between immune and cancer cells. The signaling molecules as well as the structural molecules can be readily measured using molecular profiling technology. Until now, there have not been ways to have both high-resolution imaging and high-resolution molecular profiling for the same locations in tissue. Now that the technology exists to measure both molecular profile and corresponding image, we can study the relationship between molecules in the tissue and the resultant tissue architecture in order to determine functional relationships between the two in cancer. We find that image features correspond well with extracellular matrix and other mechanisms that are important in tumor structure and aggressiveness.

**Abstract:**

Background: Cancer is the leading cause of death worldwide with breast and prostate cancer the most common among women and men, respectively. Gene expression and image features are independently prognostic of patient survival; but until the advent of spatial transcriptomics (ST), it was not possible to determine how gene expression of cells was tied to their spatial relationships (i.e., topology). Methods: We identify topology-associated genes (TAGs) that correlate with 700 image topological features (ITFs) in breast and prostate cancer ST samples. Genes and image topological features are independently clustered and correlated with each other. Themes among genes correlated with ITFs are investigated by functional enrichment analysis. Results: Overall, topology-associated genes (TAG) corresponding to extracellular matrix (ECM) and Collagen Type I Trimer gene ontology terms are common to both prostate and breast cancer. In breast cancer specifically, we identify the ZAG-PIP Complex as a TAG. In prostate cancer, we identify distinct TAGs that are enriched for GI dysmotility and the IgA immunoglobulin complex. We identified TAGs in every ST slide regardless of cancer type. Conclusions: These TAGs are enriched for ontology terms, illustrating the biological relevance to our image topology features and their potential utility in diagnostic and prognostic models.

## 1. Introduction

In the United States, cancer is one of the primary public health issues because of its high prevalence [1]. According to the World Health Organization (WHO), cancer is considered the most common cause of death across the globe, within which breast cancer and prostate cancer are the two most widespread cancer types affecting women and men, respectively [2]. Breast and prostate cancers are the most common sex-specific cancers. In 2020 alone, breast cancer affected more than 2.25 million individuals worldwide and claimed the lives of more than 680,000 women [2]. Prostate cancer affected more than 1.4 million men worldwide in 2020 [2].

In the biology of cancer, the tumor microenvironment (TME) consists of resident tissues, various inflammatory and stromal cells, and the ECM [3]. Tumors usually contain elevated levels of ECM markers [3]. Using the levels of ECM as a tumor indicator could aid in detecting cancer in its early stages and help in treatment as well [4]. The ECM has been identified as a mediator of drug resistance, therefore, attempting to modulate the ECM could improve the efficacy of cancer treatments [3]. Histopathology can be analyzed with measures of cell morphology and organization, like topological data analysis (TDA), in order to understand cellular interactions in the TME [5].

TDA [6,7] is a useful mathematical framework that assumes data points have an inherent relational structure. This makes it amenable to high-dimensional datasets like those used in biomedical research and has previously been used for image analysis, proteomics, and other omics techniques, but bioinformatic applications are relatively recent, so validated links from topological features to molecular function are still lacking [8]. Due to its efficiency in analyzing molecular profiles in cancer cells, advanced bioinformatics methods could be promising for cancer diagnosis and treatments within the field of oncology [9]. While most existing methods focus on analyzing the topology of gene expression data, our focus is on topology arising from the spatial layout of cells observed in an image (i.e., H&E-stained pathological specimens). Such image topological features have been found relevant to clinical outcomes [10,11].

To the best of our knowledge, this is the first publication linking topological features from cancer histology to spatial transcriptomics. Previously, imaging has been leveraged in spatial transcriptomics to infer gene expression at higher resolutions [12]. In prostate cancer, a convolutional neural network pre-trained on prostate needle biopsies was applied in order to extract image features from prostate spatial transcriptomic samples. These features were correlated with a factor analysis of gene expression, demonstrating how morphology colocalizes with certain gene sets in prostate spatial transcriptomic data [13]. Many methods use image features extracted from neural networks: for example, canonical correlation analysis to link bulk gene expression to embeddings from auto-encoders [14], and using a large amount of spatial transcriptomic samples to train and predict the spatial gene expression of several genes from only histopathological images [15].

In these analyses, we used correlation analysis to identify gene and ITF clusters in a similar fashion to gene co-expression module detection methods [16]. The association between gene expression and image topological features was determined by generating correlation matrices and visualization using heatmaps. Specifically, we applied methods to generate correlation matrices that allowed us to find correlations between gene expression and ITFs in breast and prostate cancer. Genes that correlate well with ITFs are called TAGs, i.e., *topology-associated genes*. Functional enrichment analysis is a useful technique for distinguishing protein and gene families related to both normal and abnormal body pathways [17]. We applied this technique to TAGs in order to find gene ontology terms that are significantly associated with ITFs from breast cancer and prostate cancer.

## 2. Materials and Methods

### 2.1. Datasets

We downloaded a total of six ST slides from the 10x Genomics Visium platform repository. Samples were either preserved by Formalin-Fixed Paraffin-Embedding (FFPE) or methanol fixation. They contain gene expression information, spatial information, and a corresponding tissue image. From these images, we extracted 1400 image topology features using topological data analysis techniques [7]. The slides that we used are FFPE Human Breast Cancer, Parent Human Breast Cancer, Parent Visium Human Breast Cancer, Human Prostate Cancer, Prostate Acinar Cell Carcinoma, and Visium FFPE Human Normal Prostate. There are between 2000 and 4000 ST spots in each sample. Breast cancer samples were taken from the breast tissue of women older than 18 years old and the slides each contained more than 4000 genes after filtering out low-expression genes. Prostate cancer spatial transcriptomic samples contained between 2543 and 4371 genes after filtering. Summary of correlation analysis and clustering results across all datasets and image feature types can be found in Appendix A.

### 2.2. Statistical and Correlation Analyses

R software was used for all statistical analyses. The correlation coefficients between image topology features and gene expression were used to identify TAGs. Of the genes with no more than 25% zero-read entries, the top 150 most variable genes in the ST data were selected for analysis. We selected 150 genes because they represent more than 50% of the total variance in the dataset, and this number was also large enough to provide statistical power in the functional enrichment software. Functional enrichment can help determine which proteins and genes have relationships with cancer [17]. We calculated the log-transform of gene expression and topology image feature values and then calculated the Pearson correlation coefficients between the ITFs and the genes. The *pheatmap* package [18] was used to visualize correlations between genes and ITFs.

### 2.3. ITF Generation

Image features were generated by a collaborating lab that specializes in computational pathology and TDA. They were extracted from 350 × 350-pixel image patches centered around the ST spots that corresponded to the same locations from which the gene expression was measured on the ST slide. The 350-pixel size was chosen to tile the image into non-overlapping squares so that features from one ST spot did not overlap with any others. We first identified cell locations from H&E images. Next, we used TDA methods to generate topological features. The cell locations can be visualized as points on a 2-dimensional scatter plot. The diameter of these points grows until they touch one another. As the diameter grows, more and more circles touch. When this happens, lines drawn from the circle centers can create shapes like triangles, squares, and rings. This is referred to as the “birth” of the shape. Eventually, the circle diameter is so large that all points are connected, drowning out shapes. The lifespan (“persistence”) of these shapes as the circles continue to grow is known as “persistent homology”. By scanning the connectedness of data at many different resolutions, persistent homology is a way to identify robust patterns in complex data from a very small to a very large scale. Persistent homology can be visualized in a persistence diagram, which can then be converted to a vector for analysis [19]. We focused on 0-dimensional and 1-dimensional topological components, which are clusters and rings, respectively.

Topology also has a distinct advantage over traditional convolutional neural networks (CNN): the features extracted by CNN kernels are task-specific, whereas topologies are not. For example, a CNN that classifies humans, dogs, and cats, may not provide features that are useful for histopathological images. Topology, instead, is a generalizable mathematical framework used to describe the shape of data and can be used to study the spatial relationships of cells on a local and global/tissue-wide level. Traditional convolutional neural networks cannot be used to process an entire whole slide image, so cannot directly represent tissue-wide architecture. Finally, because topology is not task-specific it allows us to easily compare results across cancers. The complementary functions of topology and convolutions have led to them being jointly implemented for the analysis of histopathology in cancer [20].

### 2.4. Clustering Correlated Gene Expression and Topology to Identify Sets of TAGs

As mentioned in 2.2, we identified TAGs by first calculating the Pearson correlation coefficient (PCC) between gene expression with topology across the entire spatial transcriptomics sample. To identify clusters of tightly correlated genes and topology features, we used the R package *pheatmap* [18] to identify clusters for rows (gene expression) and columns (image topology features) with a baseline of 10 clusters, then iteratively found the optimal number of clusters using the “elbow-method”. K-means and the “elbow method” are commonly used to find the optimal number of clusters [21]. The optimal number of clusters was visualized with *pheatmap*. These clusters of topology-associated genes were used for functional enrichment, providing the link between topology (i.e., tissue architecture) and the transcriptional signatures of specific biological phenomena.

### 2.5. Functional Enrichment Analysis

The R package *gprofiler2* is a useful tool for performing functional analysis of gene sets [22]. This package provides a list of enrichment terms for each gene set and corresponding *p*-values. After adjusting *p*-values using the holm method, we used a significance level of 0.05 to select the gene sets with the lowest *p*-values, which are the highest significant terms.

### 2.6. Integrative ITF Analysis in FFPE Human Breast Cancer ST Data

Based on a subset of image topology features identified through the previous correlation and functional enrichment analyses, we evaluated the relationships of the extracellular matrix ITFs with immune signaling. First, we identified a subset of 12 ITFs and 19 genes that identify ST spots related to immune signaling. This was achieved by looking at the common expression of immune-related on the ST array (e.g., major histocompatibility complex class I (MHC-I), CD8, and T-cell receptor (TCR) genes), and pathologist-defined tissue labels in order to identify ITFs most specific to these immune regions. We then used our recently developed tool *Diagnostic Evidence Gauge of Single-cells* (*DEGAS*) [23] with single-cell RNA sequencing data (scRNA-seq) from the triple-negative breast cancer (TNBC) samples [24] to estimate T-cell enrichment across the whole ST array. We also used *DEGAS* to infer ITFs in the single cells from the same TNBC dataset in order to identify which subsets of cells were more associated with specific ITFs. Finally, in the ST data, the 12 ITFs from each spot were used to train *XGBoost* [25] and *LightGBM* [26] classifiers to predict the individual expression of the 19 immune-signaling genes at that spot using leave-one-out cross-validation. The accuracy of the gene expression prediction was evaluated using the Pearson correlation coefficient between the predicted and true gene expression in the hold-out group (Appendix A).

## 3. Results

### 3.1. 1-Dimensional ITFs Correlate with Gene Sets and Functional Enrichment

We applied the workflow outlined in Figure 1. The 1-dimensional ITFs were found to have both positive and negative correlations to the 150 highest TAGs across all six of the ST slides used in this analysis (Figure 2). Furthermore, the ITFs tended to form distinct clusters presumably due to the ITFs describing similar topologies in the image. Based on the varying levels of correlation between ITF clusters and TAG clusters in the ST arrays, these similar topology ITFs can be described in terms of the associated TAG clusters. We identified several such TAG clusters in the breast cancer ST slides.

From the Parent Visium Human Breast Cancer ST slide, we found that ITF clusters 1 and 2 were more correlated with TAG cluster 6 (Figure 2A, Appendix A). ITF cluster 3 was the most similar to TAG cluster 2, representing the enrichment of the Mitochondrial inheritance, and ITF clusters 4 and 5 were the most similar to TAG cluster 4, representing the enrichment of the Collagen type I trimer (Figure 2A, Appendix A). Lastly, ITF clusters 6 and 7 were the most similar to TAG cluster 5, representing the enrichment of the ZAG-PIP complex (Figure 2A, Appendix A). In the Parent Human Breast Cancer slide, we found that ITF cluster 1 was the most similar to TAG cluster 3, representing the enrichment of the Dystroglycan binding (Figure 2B, Appendix A). ITF cluster 2 was the most similar to TAG cluster 5, and ITF clusters 4 and 5 were more correlated with TAG cluster 2, representing the enrichment of Diabetic cardiomyopathy (Figure 2B, Appendix A). Lastly, ITF cluster 5 was the most similar to TAG cluster 4, representing the enrichment of the ZAG-PIP complex (Figure 2B, Appendix A). The FFPE Human Breast Cancer slide revealed that all of the ITFs were the most similar to TAG cluster 2 (Figure 2C, Appendix A). It is also worth noting that the ECM TAG clusters were found in every one of the breast cancer ST slides and though it was not the highest ranked in terms of the PCC there were ITF clusters both positively and negatively correlated with it (Figure 2A,C). In addition to the breast cancer ST slides, there were also clear associations with TAG clusters in the prostate cancer slides.

In the Human Prostate Cancer slide, ITF clusters 1 and 2 were the most similar clusters to TAG cluster 1, representing the enrichment of the Extracellular exosome (Figure 2D, Appendix A). ITF clusters 3 and 4 were more correlated with TAG cluster 3, representing the enrichment of the Smooth Muscle Contraction (Figure 2D, Appendix A). Likewise, in the Prostate Acinar Carcinoma slide, ITF cluster 3 was the most similar to TAG cluster 1, representing the enrichment of the Extracellular exosome, and ITF clusters 1, 4, and 5 were the most similar to TAG cluster 4, representing the enrichment of the Smooth Muscle Contraction (Figure 2E, Appendix A). ITF cluster 2 was the most similar to TAG cluster 5, representing the enrichment of the Insulin-like growth factor binding (Figure 2E, Appendix A). In the normal prostate slide, Visium FFPE Human Prostate, and ITF clusters 1 and 4 were the most similar clusters to TAG cluster 4, representing the enrichment of the Smooth Muscle Contraction (Figure 2F, Appendix A). ITF cluster 2 was most correlated with TAG cluster 2, representing the enrichment of the Collagen-containing extracellular matrix, and ITF cluster 3 was most correlated with TAG cluster 1, representing the enrichment of the Extracellular exosome (Figure 2F, Appendix A).

We noted that the most significant association among all prostate ST slides was that the 1-dimensional image topology features (ITF81-ITF231) were related to the extracellular matrix. The ECM in the Human Prostate Cancer slide was gene cluster 1, which was the most similar to ITF cluster 1 (ITF1-ITF97 and ITF197-264) and ITF cluster 2 (ITF98-ITF196). In the Prostate Acinar Cell Carcinoma slide, the ECM elements were in TAG clusters 1 and 2, but only TAG cluster 1 had a high similarity with ITF cluster 3 (ITF98-ITF159). The ECM ontology terms were found in the Visium FFPE Human Normal Prostate slide TAG clusters 1, 2, and 3 as well. ITF cluster 3 was the most similar to TAG cluster 1 and ITF cluster 2 was the most similar to TAG cluster 2. Clearly, there was much overlap in 1-dimensional ITFs as they relate to the ECM.

### 3.2. 0-Dimensional ITFs Correlate with Gene Sets and Functional Enrichment

In the prostate ST slides, the 0-dimensional ITFs did not associate nearly as much with the ECM as the 1-dimensional ITFs and, instead, regularly associated with smooth muscle-related TAG clusters. The 0-dimensional ITFs showed many similarities when compared to the 1-dimensional ITFs in the breast cancer ST slides. For instance, in the breast tissue ST slides, the ZAG-PIP complex also appeared to be associated with the 0-dimensional ITF clusters.

From the Parent Visium Human Breast Cancer slide, among the most significant correlations, we found that ITF cluster 1 was the most similar to TAG cluster 3, representing the enrichment of Mitochondrial inheritance (Figure 3A, Appendix A). ITF clusters 2, 3, and 4 were most associated with TAG cluster 5, representing the enrichment of the ZAG-PIP complex (Figure 3A, Appendix A). The Parent Human Breast Cancer slide identified ITF cluster 1 as the most similar cluster to TAG cluster 3, representing the enrichment of the Dystroglycan binding. In contrast, ITF clusters 2, 3, 4, and 5 were the most correlated with TAG cluster 5, representing the ZAG-PIP complex (Figure 3B, Appendix A). In the FFPE Human Breast Cancer slide, all of the ITF clusters were the most similar to TAG cluster 2, representing the enrichment of the Centrocecal scotoma (Figure 3C, Appendix A). Unlike the similarity between 1-dimensional and 0-dimensional ITF functional assessment, the prostate cancer 0-dimensional ITFs reflected different functions than their 1-dimensional counterparts.

In the Human Prostate Cancer slide, all four ITF clusters were the most similar to TAG cluster 3, representing the enrichment of the Smooth Muscle Contraction (Figure 3D, Appendix A). In the Prostate Acinar Cell Carcinoma slide, ITF clusters 1, 2, 3, and 4 were the most similar to TAG cluster 5, representing the enrichment of the Smooth Muscle Contraction (Figure 3E, Appendix A). ITF cluster 5 was most correlated with TAG cluster 4, representing the enrichment of the IgA immunoglobulin complex (Figure 3E, Appendix A). Like the Human Prostate Cancer slide, the Visium FFPE Human Normal Prostate slide was readily associated with the smooth muscle because all of the ITF clusters were the most similar to TAG cluster 5 which represented Smooth Muscle Contraction (Figure 3F, Appendix A). In prostate ST tissues, the 1-dimensional ITFs appeared to be more informative of biology, evaluated by the number of unique TAG ontology terms, than the 0-dimensional ITFs.

### 3.3. Integrative Analysis Reveals Immune Signaling in the FFPE Human Breast Cancer Slide

Due to the high prevalence of ECM-related terms found in the preceding analyses and the lack of top rank mappings between the ITFs and the ECM TAG clusters in breast cancer ST slides, we next evaluated the subset of 1-dimensional ITFs in the FFPE Human Breast Cancer ST dataset with respect to immune signaling. A detailed look at the accompanying image to the ST slide shows that some locations in the tumor have a denser immune infiltrate than others (Figure 4A). One mechanism by which tumor cells can be detected and attacked by the immune system is through antigen presentation on MHC-I, which is subsequently detected by TCR on CD8 T-cells (Figure 4B). From this ST data, we estimated the relative abundance of T-cells using our own tool, *DEGAS* [23], and scRNA-seq from TNBC samples [24] (Figure 4C). We identified these T-cells as containing many CD8 T-cells by the abundance of CD8 in those regions (Figure 4D). The adjacent tumor region expressed high levels of MHC-I (Figure 4E) and the identified T-cells expressed higher levels of TCR genes compared to other lymphocyte-rich areas (Figure 4F). These gene expression signals seem to indicate a large number of cytotoxic CD8 T-cells attacking that specific tumor region denoted by an arrow in Figure 4. We see that the topology of the cells that are measured using ITF121 is also unique in this region, possibly denoting some arrangement of cells that influences T-cell response to the HLA-I antigen presentation (Figure 4G). We also found that when we predicted the value of ITF121 in the TNBC scRNA-seq [24] using *DEGAS* [23], epithelial cells expressing higher levels of MHC-I genes tended to have a higher association with ITF121 (*p*-value = 3.3 × 10^−5^). From these analyses, it is clear that cellular topology, measured by ITFs, is associated with different cell types and immune signaling in the TME and the ECM. The associations between tumor-infiltrating lymphocytes (TIL) and prognosis are complicated. Tumor-infiltrating lymphocytes in Ductal Carcinoma have been studied previously. High TIL densities were associated with a higher nuclear grade, more necrosis, and a higher rate of tumor recurrence. These same authors note an opposite direction of effect in triple-negative breast cancer (the most aggressive subtype) where high levels of TILs were associated with a better prognosis [27]. The topology could be applied to identify complex patterns of tumor-infiltrating lymphocytes in cancer. This is a field of active research [5,28,29].

As shown in Figure 5, we generated a correlation matrix, clustered it on the paired ITFs and TAGs using *pheatmap* (Figure 5A), then predicted the TAG expression from the ITFs by training both an AdaBoost and an *XGBoost* regression model (Appendix A). The mean AdaBoost correlation (0.524) was 0.014 greater than the *XGBoost* (0.510) during leave-one-out cross-validation but was not significant (*p*-value = 0.07, Appendix A). The ITF to TAG correlation matrix shows multiple immune-related genes (Figure 5A). Some of these genes such as HLA-E and HLA-C are directly related to MHC-I signaling by the cancer cell. The gene STAT1 is related to T-cells and immune signaling. In Figure 5B, we further performed leave-one-out cross-validation using *LightGBM* in order to predict the expression levels of each of the 19 genes based on the 12 image features. A *LightGBM* model was trained on all but one of the spots in the ST slide, reserving the last remaining spot for testing. This was repeated for each of the 19 genes. The correlations between the predicted gene expression values and the true expression values across the 19 genes ranged from 0.49 to 0.58 (Figure 5B). These results on the 10X Genomics Visium FFPE Human Breast Cancer slide showed that ITFs represent important cellular immune-tumor architectures that correspond to actual distinct molecular events and cell types and can be used to identify cell–cell interactions.

## 4. Discussion

All ST slides had sets of genes significantly correlated with topology image features. These gene sets were enriched for several ontology terms. Five ECM-related gene terms are common to the samples from both cancers: *extracellular space*, *extracellular exosome*, *extracellular vesicle*, *extracellular organelle*, and *extracellular membrane-bounded organelle*. These terms had the most significant *p*-values of the functional enrichments.

The ECM plays a significant role in tissue and organ organization and performs various functions such as cell homeostasis, differentiation, and many other physical and biochemical functions [30]. Cancer is driven by the unregulated growth of clonal cells which can evade immune system response [31]. The ECM has been shown to enhance cancer growth by changing cell functions, which can make tissues more fertile environments for cancer cells to proliferate [32]. The ECM plays a role in the progression of cancer, where the composition, toughness, hydration, and adherence to the ECM components of cells in the tumor and microenvironment evolve during breast cancer development [33]. Extracellular vesicles and extracellular exosomes can mediate cell communication and have been shown to contribute to the spread of prostate cancers [34]. Collagen plays an important part in cancer progression, sometimes contributing to the development of cancer cells and the enhancement of tumor growth [35]. Collagen Type I Trimer was significantly enriched in our analysis. Cancer-associated fibroblasts produce collagen type I on which cancer cells can migrate using fibronectin, and drugs targeting the ECM are an evolving therapeutic avenue [35].

When the FFPE Human Breast Cancer slide was studied in more detail for immune-related signaling that could impact the TME, we identified image features that appear to be predictive of immune-related genes (e.g., MHC-I). Furthermore, the localization of some image features appears to closely mirror immune signaling molecules: CD8, TCR, and MHC-I expression have a very similar distribution to the 1-dimensional ITF121. It is worth noting that ITF121 was also commonly associated with ECM in the prostate ST slides, along with ITF81-ITF231. In the Prostate Acinar Cell Carcinoma sample, we found enrichment for the IgA immunoglobulin complex. The prostate excretes IgA, and changes in IgA expression could be indicative of abnormal prostate immunity [36]. Immunoglobulins can appear at high levels in some cancers [37] and have been shown to be markers for an immunosuppressive subpopulation of B-cells in prostate cancer [38].

We identified the enrichment for the ZAG-PIP complex in the topology-associated genes from the Parent Visium Human Breast Cancer and Parent Human Breast Cancer ST slides. This is a complex of Zinc alpha2-glycoprotein (ZAG) with Prolactin inducible protein (PIP). ZAG has been studied more in prostate cancer, where this secretory protein has androgen-response enhancer elements and pushes the G1/S cell cycle transition [39]. PIP is also inducible by androgens, plays a role in the adaptive immune response, and is a marker that can be used to identify metastatic foci as having originated from breast tissue [40]. In Pip deficient mice, researchers found abnormal lymphoid proliferations in prostatic tissues and hypertrophy of the thymus medulla [41].

The chief advantage of this study is the novel use of ITFs to describe the cellular topologies associated with gene expression using spatial transcriptomics. Unlike convolutional neural networks, topological features can explicitly describe tissue architecture. We use the term tissue “architecture” because topology describes the shape of networks of cells through the study of “persistent homology” (described previously), providing a complement to measuring the morphology of individual cells which our group has studied extensively previously [42,43]. The chief limitation of our study is in not having visualized topological features for interpretation by a pathologist. This is of great interest to us, as we plan to study it using the deep-transfer-learning framework *DEGAS* [23] to identify high-risk topologies and visualize them for histopathological interpretation.

## 5. Conclusions

In this study, we took advantage of ST in order to determine how gene expression is related to cellular topology in breast and prostate cancers. We have found TAGs that are common in both cancers and some that are unique to each cancer. ECM-related enrichment terms are the most common terms in both the breast cancer and prostate cancer samples. In the prostate ST slides, 1-dimensional Image Topological Features ITF81-ITF231 were the most tightly correlated with ECM gene clusters. Among the breast cancer slides, ZAG-PIP Complex gene expression was shared. In the prostate cancer slides, enrichment for Gastrointestinal Dysmotility gene expression and IgA immunoglobin gene sets were shared across samples. The significance of this work is in demonstrating the utility of topology features in order to describe tissue architecture, and that ITFs can be correlated with gene expression. Gene sets that are correlated with ITFs are significantly enriched for biological functions. We do not contend that these biological functions are guaranteed to be present in the tissue. Instead, we introduced the important benefit of topology as means to rigorously quantify tissue architecture. Pathological assessment is usually largely qualitative (or, at best, semi-quantitative), therefore, quantification of tissue architecture is necessary in order to take advantage of histopathological information for integrative omics research. The histopathology of cancer is crucially important for the diagnosis and prognosis of patients–the amount and configuration of reactive tumor stroma, the spatial arrangement of TILs, and the density of tumor cells all have prognostic implications for patients and underlying genetic correlates. Here, we have demonstrated that topology is a promising technique to quantify the complex, high-dimensional, and highly variable characteristics of histopathological specimens, that can be effectively integrated with other biomedical data.

We can conclude that cellular topology has clear associations with the gene expression profiles from breast and prostate cancers. Understanding the biological meaning behind image features describing cellular topology can lead to improved image feature extraction methods, predictive models based on histopathological images, and integrative omics and image analysis, which may better describe the relationships between molecular events and the spatial arrangements of cells in the tumor and microenvironment.

## Figures and Tables

**Figure 1 cancers-14-04856-f001:**
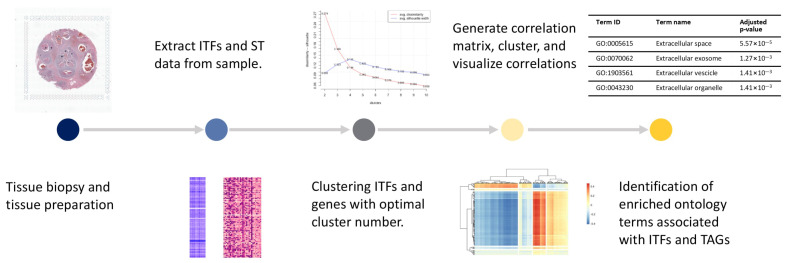
Workflow of the method from tissue preparation to identifying gene ontology terms associated with ITFs.

**Figure 2 cancers-14-04856-f002:**
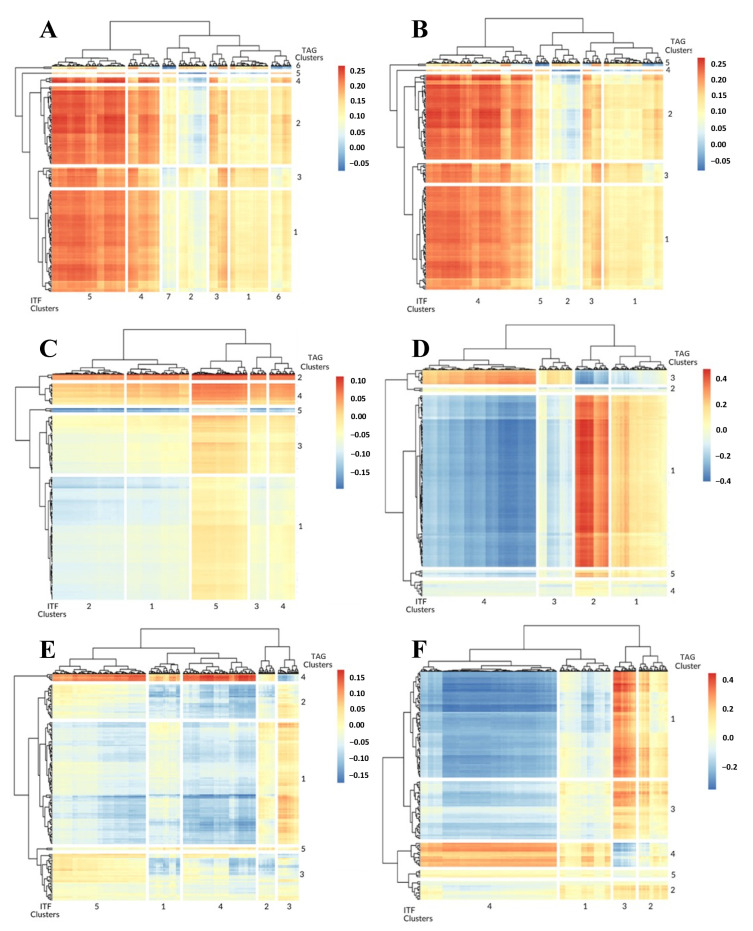
Heatmap of correlation matrix between 1-dimensional ITFs and TAG expression values for Parent Visium Human Breast Cancer (**A**), Parent Human Breast Cancer (**B**), FFPE Human Breast Cancer (**C**), Human Prostate Cancer (**D**), Prostate Acinar Carcinoma I (**E**), and Visium FFPE Human Normal Prostate (**F**).

**Figure 3 cancers-14-04856-f003:**
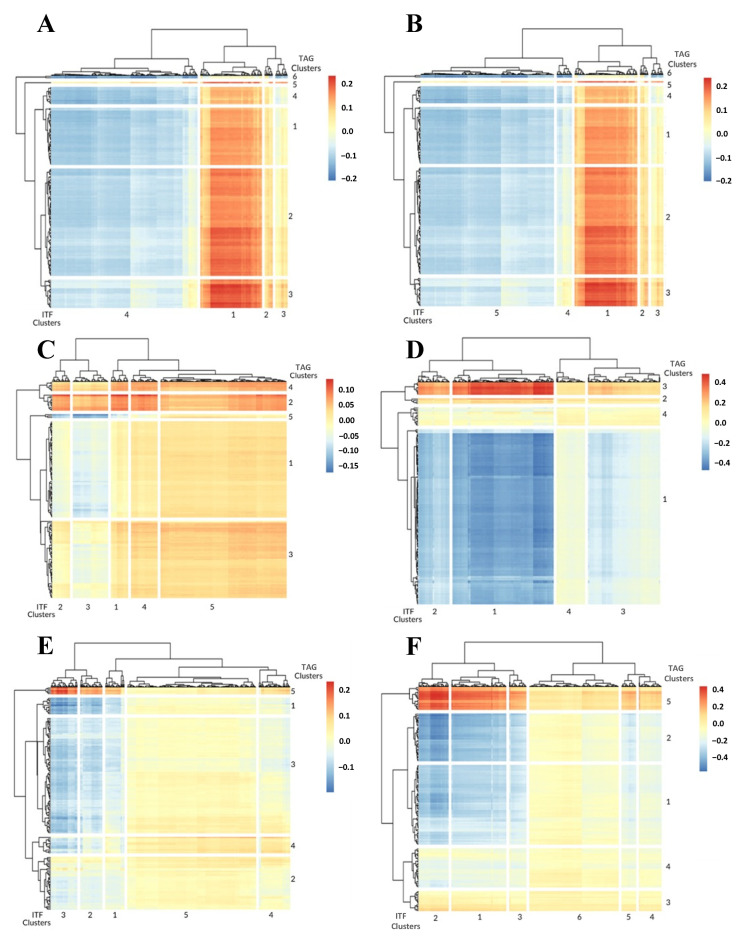
Heatmap of correlation matrix between 0-dimensional image features and gene expression values for Parent Visium Human Breast Cancer (**A**), Parent Human Breast Cancer (**B**), FFPE Human Breast Cancer (**C**), Human Prostate Cancer (**D**), Prostate Acinar CarcinoI (**E**), and Visium FFPE Human Normal Prostate (**F**).

**Figure 4 cancers-14-04856-f004:**
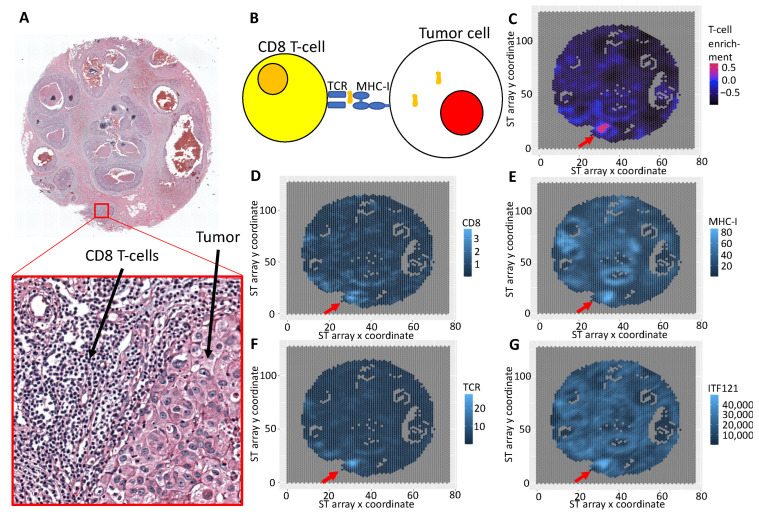
ITFs associated with cellular signaling. Here, 1-dimensional ITF121 is associated with CD8 T-cell interaction with tumor regions likely through MHC-I. Note that the periphery of the tumor region has greater ITF121. (**A**) H&E pathology image from an ST array. (**B**) Simplified diagram of T-cell and tumor signaling via MHC-I. (**C**) T-cell enrichment using DEGAS. Expression of marker genes (**D**) CD8, (**E**) MHC-I (HLA-C), and (**F**) TCR (TRAC). (**G**) ITF121 was calculated for an image patch around each ST spot. Red arrows indicate region enriched for CD8 T-cells interacting with the tumor.

**Figure 5 cancers-14-04856-f005:**
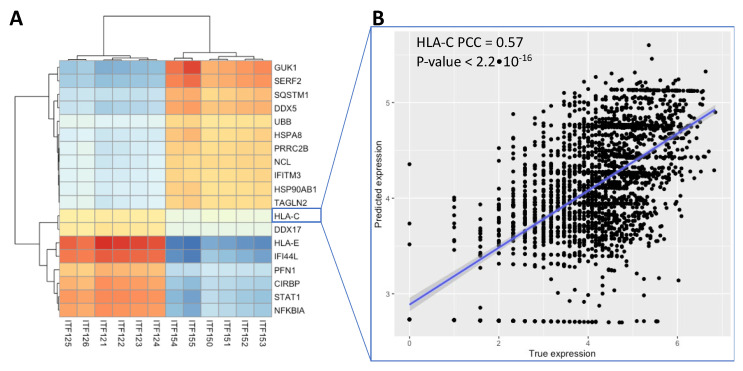
Results for predicting expression levels of TAGs using machine learning models. (**A**) Correlation between TAG expression and ITFs. (**B**) Prediction of MHC-I (HLA-C) expression using an *LightGBM* model trained on the 12 ITFs in (**A**).

## Data Availability

All of the spatial transcriptomics data and related images are freely available on the 10X Genomics data commons.

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
