# Peer review of "Spatial Transcriptomic Analysis Reveals Associations between Genes and Cellular Topology in Breast and Prostate Cancers"

_cancers, 2022, doi:10.3390/cancers14194856_

Round 1
Reviewer 1 Report
This manuscript indicated relationship of cellular topology of breat and prostate cancers with transcriptom. This is very important for understanding of cellular and molecular mechanisms of pathogenesisi of these cancers. But some corrections may be needed. In introduction, it is better to add similar studies previously reported for ST. In Materials and Methods, 2.4 ITF clustering on line 136, page 3,should be re-wriiten in detail. In 2.6. Integrative ITF analysis in FFPE human breast cancer ST data on line 164, page 4, it is better to re-write about XGboost procedure as flow-chart. In Fig.5. XGBoost regression model is used. It is better to use other algorithms such as light GBM, Catboost, adaBoost fot comparison of performance of each models. In discussion, it is better to add adgantages and limitations of this study.
Author Response
This manuscript indicated relationship of cellular topology of breat and prostate cancers with transcriptom. This is very important for understanding of cellular and molecular mechanisms of pathogenesisi of these cancers. But some corrections may be needed.
- In introduction, it is better to add similar studies previously reported for ST.
- Thank you for pointing this out to us. We have added a new paragraph to demonstrate the paucity of topology research in spatial transcriptomics, and what most existing imaging research focuses on within the field of spatial transcriptomics.
- In Materials and Methods, 2.4 ITF clustering on line 136, page 3,should be re-wriiten in detail.
- We did not notice the missing detail in the section and appreciate your suggestion. The correlation between topology features and genes, and how this is used for clustering has been further clarified.
- In 2.6. Integrative ITF analysis in FFPE human breast cancer ST data on line 164, page 4, it is better to re-write about XGboost procedure as flow-chart.
- Thank you for the suggestion. This section has been visualized in a new Figure S1.
- In Fig.5. XGBoost regression model is used. It is better to use other algorithms such as light GBM, Catboost, adaBoost fot comparison of performance of each models.
- We agree that a more thorough baseline comparison would be useful and appreciate the alternative model suggestions. We have performed a baseline comparison between XGBoost and LightGBM predicting the 19 genes from the 12 image features that we identified (Table S8). LightGBM does perform moderately better than XGBoost so we have replaced our XGBoost results with the LightGBM results (Figure 5B). The R regression implementations for CatBoost and AdaBoost either did not exist (AdaBoost) or had problems with dependencies (CatBoost). For these reasons, they were not included in the comparison.
- In discussion, it is better to add advgantages and limitations of this study.
Thank you for the suggestion. The chief advantages and limitations have been added to the Discussion.
Reviewer 2 Report
The paper relies on an original method but tries to reach conclusions that are not supported by the reported results. The simple enrichment for ontology terms cannot be considered as a proof for the biological relevance of image topology features and their potential utility in diagnostic and prognostic models. At the best it can be considered a first step hypothesis that should be confirmed with validated experimental procedures or more specific statistical analysis related to the identified biological processes. Moreover, most of the identified processes, such as the involvement of ECM, are well known by several decades in the cancer field and the description is too generic to be of interest. One of the main conclusion reported by the authors is the identification of distinct TAGs that are enriched for GI (gastrointestinal) dysmotility in prostate cancer. Of course, this sentence is too vague, has no biological meaning in itself and needs a deeper approach by using multiple morphological, cell biology and biochemical assays in order to support it.
Author Response
The paper relies on an original method but tries to reach conclusions that are not supported by the reported results. The simple enrichment for ontology terms cannot be considered as a proof for the biological relevance of image topology features and their potential utility in diagnostic and prognostic models. At the best it can be considered a first step hypothesis that should be confirmed with validated experimental procedures or more specific statistical analysis related to the identified biological processes. Moreover, most of the identified processes, such as the involvement of ECM, are well known by several decades in the cancer field and the description is too generic to be of interest. One of the main conclusion reported by the authors is the identification of distinct TAGs that are enriched for GI (gastrointestinal) dysmotility in prostate cancer. Of course, this sentence is too vague, has no biological meaning in itself and needs a deeper approach by using multiple morphological, cell biology and biochemical assays in order to support it.
- Thank you for your feedback – we have specified that conclusions found herein are associations of topological features with functionally-enriched gene modules, not some direct biochemical assay. We have also further specified that the significance of this work is demonstrating the utility of topology features with being able to correlate with gene expression, and that this gene expression is significantly enriched for known biologically functions. It is not to say that we know for sure that these biological functions are present in every spot in the tissue but rather to show the broader association between topology and gene expresion. The importance of topology is to bring a rigorous study of tissue architecture into the fold of integrative omics analysis. Tissue architecture in cancer is important for diagnosis and prognosis of patients – the amount and configuration of reactive tumor stroma, the spatial arrangement of tumor-infiltrating lymphocytes, and the density of tumor cells all have prognostic implications for patients and underlying genetic correlates. Here, we have demonstrated that topology is a promising technique to quantify these complex, high dimensional, and highly variable characteristics of histopathological specimens.
Reviewer 3 Report
In this study, Alsaleh and colleagues investigated the correlation between gene expression and histopathological features of breast and prostate cancer to identify "topology-associated genes" (TAGs).
They concluded that cell topology is associated with gene expression profiles in these types of tumors.
This is an interesting topic that highlights the potential future use of image analysis in understanding the biological characteristics of cancer to identify new targets for therapy.
However, one point need clarification.
From an immunological points of view, the authors identified a defined immune response characterized by CD8 T cells in breast and IgA in prostate cancer.
Are these specific kinds of immune responses related to the aggressiveness of cancers? Does gene expression reflect high/low aggressiveness of tumors?
Can authors discuss more these immunological findings and the benefits of these information for cancer immunotherapy?
Minor point: Please check if the abbreviations are entered in the text in the correct order.
Author Response
In this study, Alsaleh and colleagues investigated the correlation between gene expression and histopathological features of breast and prostate cancer to identify "topology-associated genes" (TAGs).
They concluded that cell topology is associated with gene expression profiles in these types of tumors.
This is an interesting topic that highlights the potential future use of image analysis in understanding the biological characteristics of cancer to identify new targets for therapy.
However, one point need clarification. From an immunological points of view, the authors identified a defined immune response characterized by CD8 T cells in breast and IgA in prostate cancer.
- Are these specific kinds of immune responses related to the aggressiveness of cancers? Does gene expression reflect high/low aggressiveness of tumors? Can authors discuss more these immunological findings and the benefits of these information for cancer immunotherapy?
- This is an excellent, and very complicated question. It depends on how you define “aggressiveness”. Fast-growing tumors can have very strong responses from the immune systems (largely from CD8+ T cells), and a good prognosis or bad prognosis. Some slow-growing tumors have very poor immune response but have a good prognosis because they are so slow-growing. The common denominator is that engaging the immune system is beneficial in the treatment of cancer, but the presence of immune cells has different prognostic implications for disease. For example: in Ductal Carcinoma in-situ (DCIS, the stage of breast cancer we analysed), a high density of tumor infiltrating lymphocytes was associated with a higher nuclear grade, more necrosis, and a higher rate of tumor recurrence. But, in triple negative breast cancer, high quantities of tumor infiltrating lymphocytes are associated with a good prognosis. The ability to leverage the immune system for the benefit of cancer treatment will revolutionize oncology, but the associations between immunological phenomena in histopathology and prognosis are myriad. Previously, our research group has applied spatial statistics to describe tumor infiltrating lymphocytes and found correlates with prognosis. Topology could also be applied to study tumor immunology, but we did not specifically analyze lymphocyte infiltration topology in this paper. I have summarized the above and added citations for readers interested in this topic.
- Minor point: Please check if the abbreviations are entered in the text in the correct order.
- Thank you again for your comment. We have gone through the paper multiple times to check the abbreviations and ensure that they are defined at first occurrence. We corrected many during this process.
Round 2
Reviewer 1 Report
This manuscript was corrected clearly according to reviewer's comments. Therefore, it will be significant to the research fields.
Reviewer 2 Report
The authors added some sentences that point out some limitations of the study and provide a more balanced view of strengths and weaknesses of the obtained results. The reader can now get the originality of this approach without overstatements about the present stage of the research.